# Cleaner fish are sensitive to what their partners can and cannot see

Katherine McAuliffe [1✉], Lindsey A. Drayton[2], Amanda Royka[2], Mélisande Aellen [3], Laurie R. Santos[2] & Redouan Bshary [3]

Much of human experience is informed by our ability to attribute mental states to others, a capacity known as theory of mind. While evidence for theory of mind in animals to date has largely been restricted to primates and other large-brained species, the use of ecologically-valid competitive contexts hints that ecological pressures for strategic deception may give rise to components of theory of mind abilities in distantly-related taxonomic groups. In line with this hypothesis, we show that cleaner wrasse (*Labroides dimidiatus*) exhibit theory of mind capacities akin to those observed in primates in the context of their cooperative cleaning mutualism. These results suggest that ecological pressures for strategic deception can drive human-like cognitive abilities even in very distantly related species.

[1] Department of Psychology, Boston College, Chestnut Hill, MA, USA. [2] Department of Psychology, Yale University, New Haven, CT, USA. [3] Department of Behavioural Ecology, Institute of Biology, University of Neuchâtel, Neuchâtel, Switzerland. ✉email: mcaulikg@bc.edu

The ability to represent others' perceptual states in strategic interactions is a core component of human as well as non-human primate theory of mind (ToM) abilities. Comparative research has shown that non-human primates possess some key components of ToM, such as the ability to represent what others perceive and know[1–5]. Indeed, many studies of ToM in non-human primates make use of food competition situations to test what primates know about others' perceptions. For example, rhesus monkeys steal more often from a human competitor whose face is hidden by an opaque barrier than a competitor whose body alone is hidden[6]. Similarly, subordinate chimpanzees selectively take food that is hidden from a dominant competitor's view behind an opaque barrier over food placed in front of a barrier so the competitor could see it[7]. Non-human primates can also track what a competitor has seen in the recent past[8,9] and can strategically conceal visual[6,10,11] and auditory[12,13] information in an attempt to deceive their rivals. While the exact mechanisms remain debated (see, e.g.[14–16]), these results and others (see reviews in[17,18]) demonstrate that non-human primates share one of the fundamental hallmarks of our human ToM reasoning: the ability to represent what others see and know.

To date, evidence for tracking what others see has been restricted to closely-related species such as non-human primates and other species who share a primate-like brain-to-body size ratio (e.g., corvids:[19,20]). This pattern raises the question of whether large brains are a prerequisite for ToM components or whether selective pressures for successful deception may shape the evolution of ToM capacities even in smaller-brained species, like ectotherm vertebrates.

The mutualistic cleaning interactions of the bluestreak cleaner wrasse (Labroides dimidiatus; hereafter cleaners) present a unique opportunity to examine this question. Client fish approach cleaners in order to be cleaned of ectoparasites and dead skin cells, but cleaners prefer to feed on client mucus, where the consumption of the latter constitutes cheating[21]. Cheating often causes clients to terminate the interaction prematurely[22]. As a consequence, when pairs of male and female cleaners co-inspect a client, they face a situation akin to a prisoner's dilemma: cheating before the partner does yields the benefits of mucus consumption, while both partners bare the cost of the client leaving[23]. Cleaner pairs solve the dilemma but largely in an asymmetric way: the larger males punish a cheating female partner with aggressive behavior such as chasing and biting, causing her to behave on average very cooperatively during joint inspections[24]. Previous work has shown that cleaners possess a surprisingly sophisticated understanding of this social situation and that they vary their levels of cooperation depending on their specific social context[25,26]. For example, cleaners are sensitive to the presence of bystander client fish; cleaners cooperate more with their current client fish when they are observed by a bystander client fish than when the cleaners are unobserved[25,26]. The existence of such audience effects in this species is consistent with the possibility that cleaners may track what others around them can and cannot see and may choose to cheat more often when others are not perceptually aware of their actions.

To test whether cleaners may use representations of others' perceptions to cheat more effectively, we took advantage of the fact that male cleaners punish female partners who cheat[23,24,27]. This feature of the cleaner fish system means that female cleaners may have a motivation to cheat more often—that is, eat more of the high-valued client mucus—when their male partner cannot see them and thus would not know to punish them. We hence adapted procedures more typically used to test primate ToM[7] for use with cleaners. By varying male visual access to female foraging behavior in two experiments, we tested whether females indeed adjust their behavior as predicted if they know what males can and cannot see.

In Study 1, we varied whether males could or could not observe their female partners "cleaning" artificial client fish (a plexiglass plate) in a separate compartment (see Fig. 1). Each plate contained 12 less-preferred flake items and 2 preferred prawn items and remained submerged as long as the female ate against her preference (i.e. eating flake items equated to "cooperation") but was removed as soon as she ate according to her preference (i.e. eating a prawn item equated to "cheating"). After a cheating event, the partition separating the male was removed and we monitored punishment behavior for approximately 30 s. If males as bystanders punish females for cheating and if female cleaners track the perceptual access of their male partners, then females should cheat more often when their partner is stationed behind an opaque partition than when he is behind a transparent partition.

## Results and discussion

Our measure of interest was whether females ate more of the flake items (i.e., cheated less quickly) when the male had perceptual access than when he did not. Consistent with our predictions, females ate fewer flake items in the male not visible condition (Mean = 3.6 items, SD = 3.2) than in the male visible condition (Mean = 4.5, SD = 3.1; Fig. 2A; for flake items eaten by pairs see Supplementary Fig. S1). Additionally, females tended to cheat less over rounds (Supplementary Fig. S2). Indeed, the number of flake items eaten was predicted by both conditions (LRT, $X^2_1 = 8.13$, $p = 0.004$; Fig. 2A; Supplementary Table S1) and round (LRT, $X^2_1 = 12.46$, $p < 0.001$). In a separate model, we found that the interaction between condition and round was not a significant predictor (LRT, $X^2_1 = 0.25$, $p = 0.6$). Thus, like non-human primates[6,7], cleaners cheat more when others cannot see them.

We also explored the role of punishment in this effect. We found that punishment did not vary as a function of the interaction between condition and female cooperation, measured as the number of flake items eaten (LRT, $X^2_1 = 0.11$, $p = 0.7$) and indeed the full model containing this term was no better than the null. However, when this interaction term was removed, we found that, consistent with past work[24], punishment was negatively related to female cooperation. Regardless of whether or not they could see the female, males were less likely to punish the more flake items females ate (LRT, $X^2_1 = 4.18$, $p = 0.04$; Supplementary Table S1; Supplementary Fig. S3). That males punished regardless of whether they had visual access to females' eating suggests that they may punish not merely in response to seeing a female cheat but in response to some aspect of females' behavior following cheating.

To further unpack the relationship between males' punishment and female cheating, we looked for individual differences in females' propensity to cheat based on male perceptual access. To this end, we created a "strategic cheating score" by subtracting the flake items eaten in the male not visible condition from those eaten in the "male visible" condition. This measure revealed that the majority of females cooperated more in the perceptual access, male visible, condition than in the no perceptual access, male not visible, condition (Fig. 2B), suggesting the condition difference reported above appears at the individual-level as well (see also Supplementary Fig. S1 for individual data). The strategic cheating score also allowed us to test whether females that cheated more often when unobserved were those who benefited most from such adjustment—namely those paired with a more punitive male. Males naturally vary in how much they punish females[27] and it would, therefore, benefit individual females to be more strategic

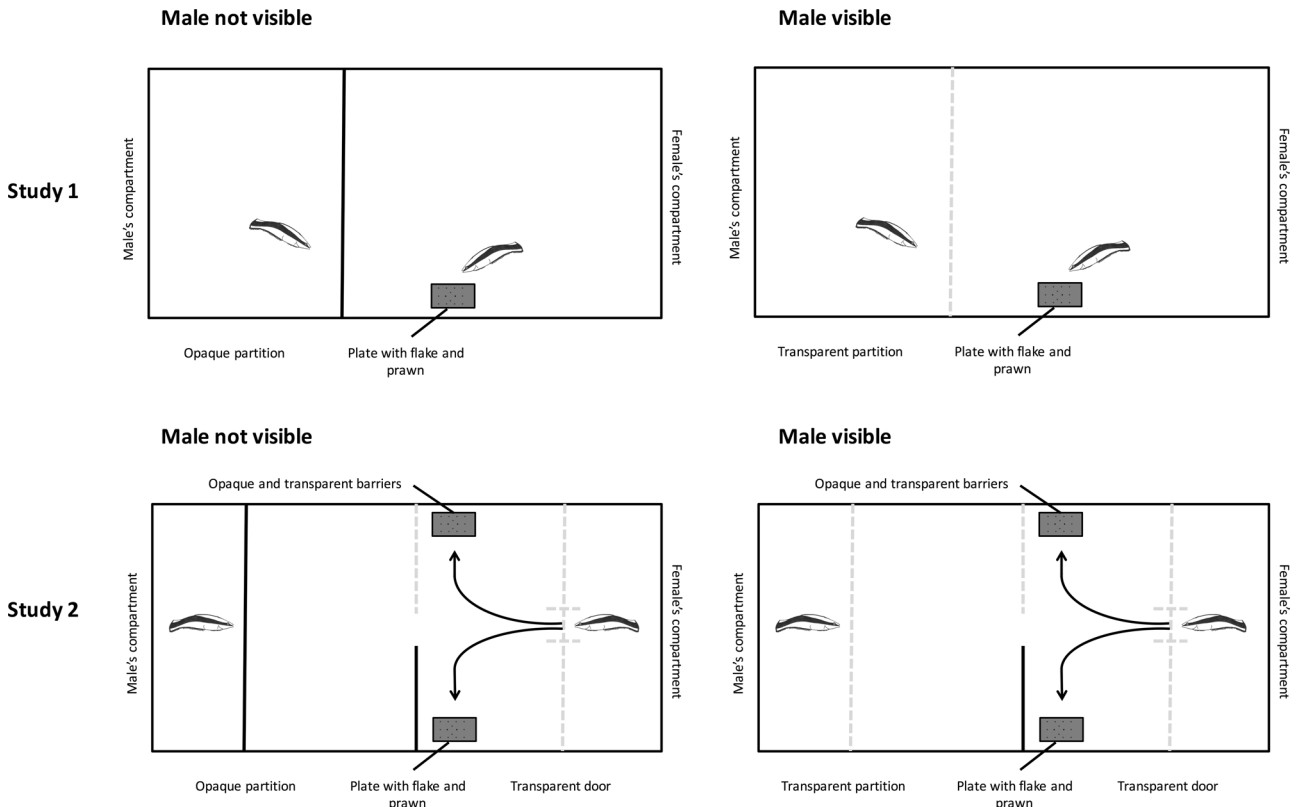

**Fig. 1 Diagram of experimental set up for Studies 1 and 2.** Diagram of the experimental set up for Study 1 and 2. In Study 1, females and males were separated by either an opaque (male not visible condition) or transparent partition (male visible condition). The female was presented with a plate with 12 flake and 2 prawn items. The plate was removed when she ate a prawn item and the male was released from his compartment and punishment was monitored. In Study 2, males were again separated by either an opaque or transparent partition. The female began each trial behind a transparent partition with a small door at its center. Plates with 12 flake and 2 prawn items were presented behind an opaque and transparent barrier. When the plates were in place, the door was lifted and the female swam through the center of the choice arena. The female then chose to eat at the plate behind one of the barriers and the other plate was removed. The chosen plate was removed when she ate a prawn and the male was released from his compartment and punishment was monitored.

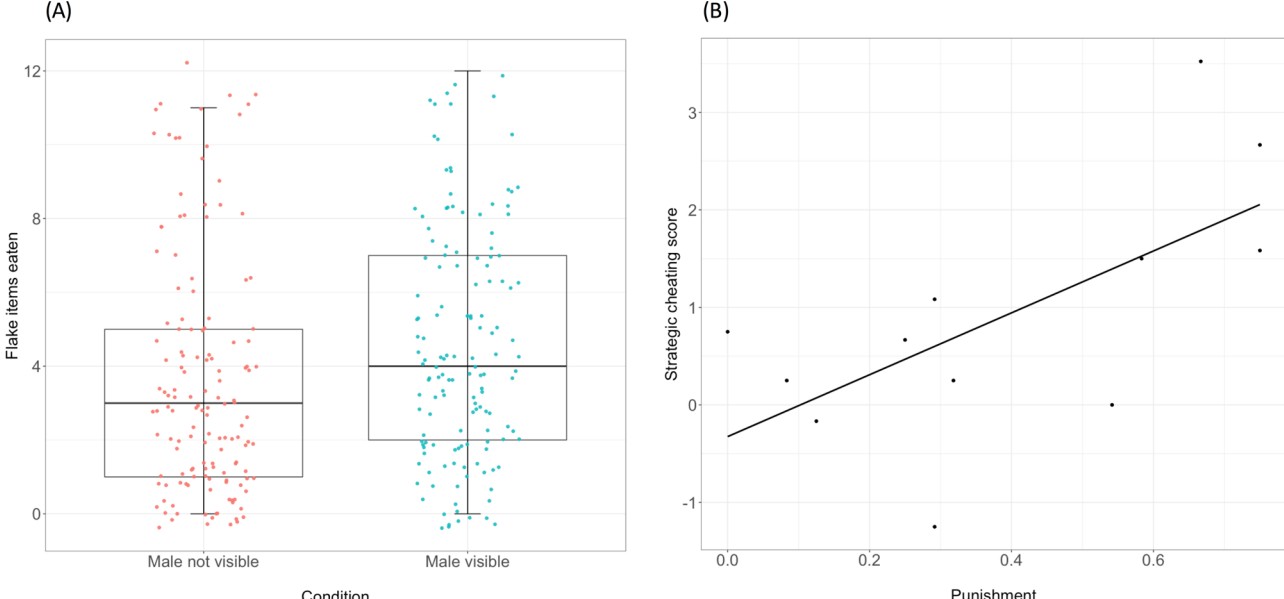

**Fig. 2 Flake items eaten by females and female strategic cheating scores from Study 1.** Results from Study 1 ($N = 12$ pairs of fish). **A** Boxplot showing flake items eaten by females across conditions along with raw data (For flake items eaten by pairs see Supplementary Fig. S1). Box shows the interquartile range, the horizontal line shows the median, the whiskers stretch from the minimum to maximum and raw data are overlaid as dots colored by condition. **B** Plot showing the relationship between female strategic cheating score and male punishment.

(i.e., to cheat more behind the opaque barrier) when paired with more punitive males. As predicted, our female strategic cheating measure correlated with male punishment ($N = 12$, $r = 0.64$, $p = 0.02$; Fig. 2B). With the caveat that one can never be certain of the directionality of a correlational effect, these data hint that females who experienced more punishment from males showed more sensitivity in their cooperative behavior depending on whether or not they were being watched than females that did not experience much punishment.

Like primates (see ref. [6]), female cleaners cheat more when their partners lack perceptual access to their behavior. Males apparently responded to female foraging behavior, being less likely to punish if the female ate against preference. Female cleaner fish, therefore, seem to take into account males' visual access and are more cooperative when they are being observed than when they are hidden behind an opaque partition. Nevertheless, there is one potentially important difference between our experimental design and standard designs for primates: we could not construct a barrier that would allow the females to always see the male body but allow us to vary whether or not the male can see the female. Therefore, the results of the first experiments could be at least in part due to females responding to the presence/absence of the male partner. However, as female strategic behavior is correlated with male punishment—likely because females paired with more punitive male showing a more pronounced response to barrier condition—we propose that females' performance is not merely an automatic reaction to being behind an opaque or transparent partition.

Study 2 used a more complicated choice task to explore whether female cleaner fish take into account a male's perceptual access when *planning* where to feed. Females were given a choice to feed behind a transparent or opaque barrier while their males were separated. In one condition, males were separated by a transparent partition (male visible), and thus the female's choice mattered for the males' perceptual access (Fig. 1). In a second condition, males were separated by an opaque barrier (male not visible); note that in this condition females should be indifferent in their choice of the transparent and opaque partitions since the males' perceptual access was already blocked by the opaque barrier. Unlike Study 1, in Study 2 rounds were blocked such that each condition was presented once across six trials within the first block of 12 trials and then again in the second block of 12 trials— i.e., fish saw both conditions presented in block 1 and then again in block 2.

We first examined females' choices across condition and block. Condition was not a significant predictor of choices (LRT, $X^2_1 = 0.003$, $p = 0.954$) whereas block was (LRT, $X^2_1 = 3.66$, $p = 0.056$; Supplementary Table S2). However, this model was no better than the null (Supplementary Table S2). In a second model, we found that the interaction between condition and block was a significant predictor of females' choices (LRT, $X^2_1 = 22.13$, $p < 0.001$). As shown in Fig. 3, when females first encountered the experimental conditions in block 1, their behavior conformed to our predictions: they were more likely to feed behind the opaque barrier when the male was visible. However, this pattern of behavior reversed in the second block, in which females were more likely to choose to feed behind the transparent barrier when the male was visible (see Supplementary Fig. S4 for data separated by pair). To better understand females' behavior in this task, we next turn to analyses of male punishment behavior and female feeding behavior.

We first examined whether male punishment varied as a function of condition, females' choices and the interaction between the two terms in a GLMM. We found no effect of the interaction (LRT, $X^2_1 = 0.51$, $p = 0.477$). When this term was removed, we found that males were more likely to punish—

regardless of condition—when females chose to feed behind the opaque barrier (LRT, $X^2_1 = 4.48$, $p = 0.034$; Supplementary Table S2). This result represents an interesting convergence with the male punishment results from Study 1. It is possible that females behave differently after they make a choice (e.g., give off a behavioral cue depending on whether they chose to feed in private), and that males adjust their level of punishment based on those cues. The fact that males punished more when females chose to feed behind the opaque barrier could explain why some females shifted their choices shifted towards feeding behind the transparent barrier between blocks 1 and 2.

Female cooperation, measured in terms of the number of flake items eaten prior to eating a prawn item, varied marginally as a function of their barrier choice and condition (LRT, 2-way interaction, $X^2_1 = 3.51$, $p = 0.061$; Supplementary Table S2). This marginal effect is likely spurious, as the model including this interaction term was not a better fit to the data than the null. A significant two-way interaction would have been consistent with the prediction that females assumed that they could get away with cheating behind barriers in the male visible condition. However, when we examine this relationship in the first block alone—where females showed a preference for the opaque barrier—we no longer see an effect ($X^2_1 = 0.99$, $p = 0.32$; Supplementary Table S2). Nevertheless, this negative result does not necessarily support the hypothesis that females lack an understanding that cheating would be more "permissible" behind barriers when their male partner was visible. In contrast to Study 1—where females cheated more when the male partner could not see—male partners counteracted that logic by being more likely to punish if the female ate behind the opaque barrier, independently of condition. Thus, while females may initially base their decisions on what males can or cannot see, they may still adjust their behavior based on males' actual responses.

Taken together, these results suggest that cleaner fish show several of the hallmarks of primate ToM capacities. Like primates[6,10,11], female cleaners strategically use visual barriers to hide their actions. In addition, females use this strategic cheating strategy more often at the individual level when paired with more punitive males. Finally, first block performance on Study 2 showed that females preemptively sought opaque barriers more often than transparent barriers in cases where males could see their actions. However, this effect was fragile and disappeared after repeated trials.

One intriguing finding from both Study 1 and Study 2 was that male punishment did not depend on whether they could see females: In Study 1, male punishment was related to female cheating, regardless of their visual access. In Study 2, males punished more whenever females chose to feed behind the opaque barrier. This surprising set of findings hints that male punishment may be based on factors other than female's cheating behavior. It is possible, for example, that females may behave differently after making different kinds of choices (e.g., when they chose to feed in private vs public), and that males adjust their level of punishment based on those cues. It would be interesting to explore whether this is the case, and also to investigate whether females attend to similar male behavioral cues that may guide their decisions about how much and when to cheat. Understanding whether such behavioral cues exist and how they influence male punishment and females' decisions is an exciting avenue for future work.

If females provide cues about their cheating, then why would our wild-caught females show any sensitivity to our experimental manipulations? One potential explanation for this pattern involves a difference between our task and the situation that cleaners experience in the wild. For instance, it is possible that such cues are short-lasting under natural conditions where

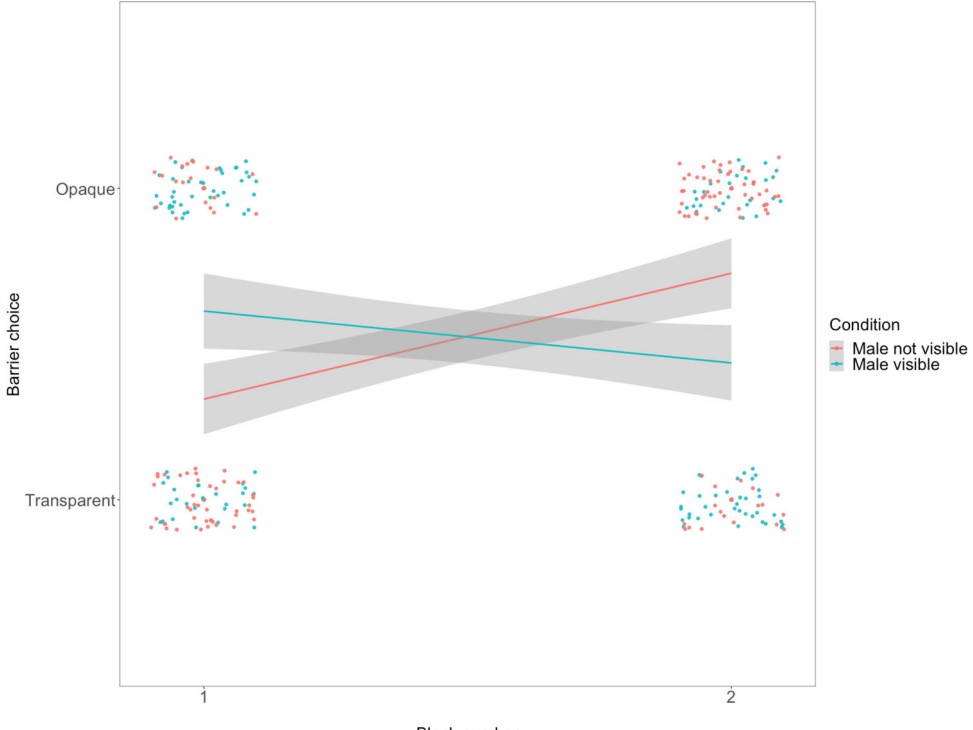

**Fig. 3 Opaque barrier choices by females in Study 2.** Scatter plot of females' choices to feed behind the opaque barrier as a function of round with plotted lines from a linear model separated by condition ($N = 11$ pairs of fish). Gray shaded area shows confidence bands.

partners may stay out of sight for prolonged periods and both males and females are often distracted by new clients demanding service. Under these conditions, any cues associated with female choices may not be noticed by males. By contrast, our studies allowed males to reunite with their females immediately after a trial with no distractions. Thus, females may attend to what males can and cannot see under natural conditions and may have demonstrated this pattern of performance—at least initially—in our experiments.

Another possibility worth considering is that female cleaners developed their ability to represent others' perspectives in a totally different context, such as when represented what image-scoring bystander clients can and cannot see[25,26]. Cleaners are highly sensitive to how bystander clients respond to their treatment of current clients[28–30]. Therefore, paying attention to whether or not bystander clients can see how cleaners treat current clients might be the primary selective force on perspective-taking abilities, which are then also applied to male cleaners.

In conclusion, our results suggest that cleaner wrasse can detect what their partners can and cannot see—one important key feature of human-like ToM capacities—in the context of their cooperative cleaning mutualism. These findings support the view that ecological pressures for strategic deception can drive surprisingly complex human-like cognitive abilities even in a distantly related fish species.

## Methods

**Subjects.** Both studies were conducted in March–April 2016 at Gump Field Station on the island of Mo'orea in French Polynesia. Twelve pairs of cleaner fish (see Supplementary Table S3 for sizes) were caught with hand nets and a barrier net (2 m long, 1.5 m high of a mesh size of 5 mm) in the surrounding water/reef by divers and brought into large round aquaria, built from plastic tank (with a diameter of 2.12 m and 0.865 m high) subdivided into four compartments with plexiglass panels. Therefore, we ended up with twelve sections of an approximate dimension of $100 \times 30$ cm$^2$, each of which housed one male-female pair. Aquaria received fresh sea water through a "flow through" system that regularly replaced

water. The water temperature was thus similar to the temperature of the surrounding water. The water reached approximately 35 cm high in each section. The tanks were outside and thus fish were subject to natural light conditions. Each tank was covered with mesh to protect the fish from birds. Each section contained two PVC pipe tubes which served as refuges for the fish. Tanks were cleaned regularly by scraping off algae and siphoning out dirty water. Fish were fed ad libitum with mashed prawn smeared on plexiglass plates every non-experimental day. At the end of the field season, fish were returned to the reefs where they were initially caught.

**Ethics.** This research was approved by the Yale and Boston College IACUCs (Protocols 2015-11627 and 2016-006-01, respectively) and by the French Polynesian authorities responsible for the program "Délégation à la Recherche". The authors declare no competing interests.

**Training.** All pairs were given a minimum of 2–3 days to acclimate to laboratory conditions prior to training. Every morning prior to an experiment, fish were given a plate with food (prawn or flake depending on the study) for five minutes followed by a break of at least 15 min before testing. Training took place in two waves: (1) training fish that the plate would be removed if they ate the high-value food item i.e. eating a prawn item, and (2) training fish that punishment may occur following the removal of the plate. We worked with six pairs at a time. We trained our second set of six pairs a week after starting training with our first set of pairs.

In the first wave, males and females were separated and given a plate ($7 \times 10$ cm) containing 12 flake (low value, unpreferred food, simulating ectoparasites) and 2 prawn (high value, preferred food, simulating client mucus) items[31] items. Flake items consisted of a small drop of fish flake mixed with mashed prawn while prawn items consisted of a drop of mashed prawn alone. The male and female had the opportunity to eat as many flake items they wanted, but as soon as they ate the first prawn item, the plate was removed quickly to simulate a client leaving. The plate was then presented again 30–60 s after removal, and the same method was followed for the second prawn item. If the second prawn item was not eaten (with the subject eating flake items or nothing at all), we waited five minutes and then removed the plate. For subjects that had not eaten any flake items during the two plate presentations, we introduced a plate with 14 flake items to make sure that these subjects were not generally avoiding flake items. Furthermore, such trials gave subjects the experience that eating flake items did not lead to the removal of a plate.

Each fish was presented with six rounds of this training so that they would have 12 opportunities to experience plate removal in response to eating prawn. In cases in which a subject refused to eat flake items, a plate with 14 flake items was presented. In the second wave of training, males and females remained in one compartment together and were presented with one plate containing 12 flake items and 2 prawn items. When either fish ate a prawn item, the plate was removed and

we monitored punishment behavior for approximately 30 s. Each pair received 18 rounds of this training. Between these 18 training rounds and Study 1, all pairs participated in a pilot study for another project[32]. This pilot lasted 1 day and a half and involved plates with flake and prawn items as well as opaque and clear partitions. Consequently, all individuals had extra training on the discrimination of flake and prawn items and the consequences of eating one of the two items. Given that this extra experience was invariable across subjects, it should not have affected the current study.

**Study 1 procedure**. Prior to commencing Study 1, we attempted to habituate fish to the barriers that would be used in Study 2 (see habituation below). We ceased habituation when it was clear that the fish were not comfortable with the new material in their aquaria and commenced Study 1.

For Study 1, we first presented pairs with two warm-up trials. In the first warm-up trial, pairs were separated by a clear partition and both individuals were presented with a plate with 12 flake items and 2 prawn items. Plates were removed quickly once one of the prawn items was eaten. In the second warm-up trials, males and females were in the same enclosure and, again, a plate with 12 flake items and 2 prawn items was presented. We removed the plate quickly once one of the prawn items was eaten and we then monitored punishment behavior for approximately 30 s.

For the test trials, we again separated the male and female. We gave females approximately 30 s to habituate to the separation and then presented them with plates containing 12 flake items and 2 prawn items. If the female cheated and ate a prawn item, we immediately removed the plate and the partition separating her from the male. Because males were getting less food than females in this task, we fed the males with a flake plate every two rounds.

Fish were presented with 24 trials total, 12 of which were in the male not visible condition and 12 of which were in the male visible condition. Trials were counterbalanced within blocks of 6 such that each block of 6 trials contained 3 trials in the male not visible condition and 3 trials in the male visible condition. See Supplementary Movie 1 for an example trial.

**Study 2 procedure**. Our procedure for Study 2 can be separated into three phases: habituation, warm-ups and test trials. One of the pairs was not tested in Study 2 (pair G) because the female did not habituate to the testing set up.

In the habituation phase, fish were introduced to the opaque and transparent choice barriers. The choice barriers were secured to the side of the tank with clothing pins. The night before the first test, the two barriers were set up and left overnight. All pairs completed four blocks of six trials. Condition alternated with block such that pairs were tested in one of two sequences: (1) male visible (block one), male not visible (block two), male visible (block three), male not visible (block four) or (2) the opposite order. Note that in our analyses we divide the data into two blocks, with block one containing data from the first presentation of each condition (12 trials total, 6 in each condition) and block 2 containing data from the second presentation of each condition (again, 12 trials total, 6 in each condition). Please see Supplementary Fig. S5 for data separated by four blocks of 6 trials each. We varied the sides on which the opaque and transparent choice barriers were placed across the four blocks such that pairs were tested in one of the following configurations with respect to the placement of the opaque choice barrier (right = R; left = L): opaque on RLLR, LRRL, RRLL or LLRR. The choice barriers remained in the same position for the entire day of testing. When the sides were switched, fish had a night to habituate to the new configuration.

Before test trials, fish were presented with warm-up trials. On the first warm-up, the male and female were separated by a partition. Males were presented with a plate that had four flake items and two prawn items (4F:2P). Females were gently moved to one side of their enclosures by a partition ("transparent door" in Fig. 1) with a small opening that could be open or shut. Once behind the door, the 4F:2P was lowered and the small door was opened such that females had to swim through the opening to access the plate. Plates were immediately removed if fish ate the prawn items. These warm-ups were presented twice in succession for both the males and females. In the second warm-up trial, the male and female were together and were presented with a 4F:2P plate together. As soon as a prawn item was eaten, the plate was removed.

In the test phase, males and females were separated by either a transparent or opaque partition, depending on condition. The female was moved behind the transparent door and given approximately 30 s to habituate. Once the female was behind the door, two 4F:2P plates were submerged behind the opaque and transparent barriers. This was done simultaneously in almost all cases. For two of our pairs, the doors preventing females from entering the choice compartment were not functioning properly so they needed to be held down by the experimenter to prevent the female from escaping. In these cases, one plate was lowered into the water immediately before the other one. Females were given approximately two minutes to eat the prawn item. If she did not eat the prawn within that time, the plate was gradually removed and the male was released and punishment was monitored for approximately 30 s.

Fish were presented with 24 trials total, 12 of which were in the male not visible condition and 12 of which were in the male visible condition. The conditions were presented in blocks of 6 trials and counterbalanced across pairs so that half our pairs started with the male not visible condition and half started with the male

visible condition (though note that this was ultimately slightly imbalanced because one pair did not finish the study). See Supplementary Movie 2 for an example trial.

**Coding and analysis**. Our main measures of interest were (1) the number of flake items females ate before "cheating" and eating a prawn item, which resulted in plate removal and (2) whether males punished (binary: punishment = 1, no punishment = 0). In Study 2 we were additionally interested in which barrier females chose to eat behind (binary: opaque = 1, transparent = 0). Experimenters served as live coders for our two main dependent measures: namely, how many flake items were eaten before cheating occurred and whether punishment occurred within the window following partition removal (approximately 30 s, timed using a stopwatch). We checked the reliability of our flake item DV in two ways. First, for the first six pairs tested, a researcher who was blind to condition counted the flake that had been eaten from the plates once they had been removed from the aquaria. Agreement between their counts and the live coding was very high (Pearson correlation, $r = 0.96$, $p < 0.001$, $N = 141$ trials). Second, a research assistant who was not involved in data collection reviewed videos and assessed how many flake items were eaten. Agreement between video coding and live coding was again high (Pearson correlation, $r = 0.92$, $p < 0.001$, $N = 242$ trials). A live coder recorded whether punishment occurred and endeavored to count individual punishment events. Because punishment frequently occurred behind our camera, we did not use video coding as a means of calculating reliability. Note that experimenters endeavored to count punishment events but because we could not check videos for reliability, here we report punishment as a binary variable (1 = occurred; 0 = did not occur). All analyses are based on live coded data. In cases in which there was ambiguity in our live coding (e.g., one or two flakes eaten), we systematically defaulted to the lower number. We additionally attempted to code latency to female's eating but due to poor visibility in some videos we were not able to code these data for all trials and thus do not explore latency data here. For Study 1, we are missing flake data for one trial and punishment data for two trials. For Study 2, we are missing choice data for one trial and flake data for two trials.

**Statistics and reproducibility**. All statistical models were conducted in R version 3.6.3[33]. We used lme4[34] to fit mixed models with pair identity fit as random intercepts to examine flake items eaten (linear mixed models; LMMs), presence or absence of punishment (generalized linear mixed models; GLMMs). For all dependent measures, we first created a null model which included just our random effect of pair identity and compared the null model to a full model with predictors of interest using the "ANOVA" command. Unless otherwise specified, full models provided a better fit to the data than null models. We then examined individual terms of interest by dropping them from our full model and testing whether their inclusion improves model fit using Likelihood Ratio Tests (LRTs, performed using the command "drop1"). Our primary predictor of interest was condition (male not visible, male visible). We additionally examined the effects of round (Study 1) and block (Study 2). For Study 1 analyses, round was a continuous predictor ranging from 1–12 (recall that there were 24 trials total, 12 of which were in the male not visible condition and 12 of which were in the male visible condition). For Study 2 analyses, block was treated as a two-level factor with block 1 containing the data from the first 6 trials of the male not visible condition and the first 6 trials of the male visible condition (12 trials total) and block 2 containing the second 6 trials of each condition (12 trials total).

**Reporting summary**. Further information on research design is available in the Nature Research Reporting Summary linked to this article.

## Data availability
Raw data have been uploaded as supplementary material. See Supplementary Data 1 and Supplementary Data 2.

## Code availability
The R code used for analyses is available from the corresponding author upon reasonable request.

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

## Acknowledgements

The authors thank the Gump Field Station, Martin Jeanmonod, Nichola Raihani, Swiss Science Foundation grants (31003A_153067/1 and 310030B_173334/1 to R.B), and Yale University.

## Author contributions

K.M., L.D., M.A., L.S. and R.B. designed the study; K.M., L.D., M.A. and L.S. collected the data; K.M. led analyses, all authors (K.M., L.D., A.R., M.A., L.S. and R.B.) contributed to interpreting results and writing and revising the manuscript.

## Competing interests
The authors declare no competing interests.
