## [Transparent Peer Review File · Communications Biology]

Reviewers' comments:

Reviewer #1 (Remarks to the Author):

This excellent manuscript describes an exciting study showing clear evidence for theory of mind in fishes. This is the first demonstration of this phenomenon outside of non-human primates and provides compelling evidence for a complex cognitive ability in cleaner fish. The study is very well executed and shows that paired male and female cleaner fish exhibit a complicated relationship where the female will cheat when the male is not watching her and will alter her behaviour when the males is visible. In response to cheating the male punishes the female and the degree of punishment modulates female cheating. In a second experiment female cleaner fish choose an opaque barrier to cheat behind rather than a transparent one so the male cannot see her cheating. In response to this it appears that the male punishes the female if she chooses the opaque barrier and so again this influences the future behavioural choices of the female. This is a clear demonstration of theory of mind where the female is aware of being watched and alters her behaviour accordingly. Decisions to punish females going out of site is a remarkable finding and suggests a complex relationship between the controlling male and the cheating female. These novel results are fascinating and are highly likely to spark media interest. I highly recommend publication after a few minor comments:

L85 you mention study 1 but not study 2 in this paragraph - please amend.

L247 More details are need for reproducibility reasons - what size were the animals? Presume you used natural seawater? Can you add how much water was replaced per day or week, aeration, the light dark regime, temperature and water quality parameters. Did you have ethical permission and what was the fate of the fish?

SI: would it be possible to add some videos for study 1 and 2 here?

Lynne Sneddon

Reviewer #2 (Remarks to the Author):

The authors present an exciting, well-designed, and highly novel study on the social cognitive abilities of a species of fish. As the authors note, little is known about the perspective-taking abilities of animals besides primates and corvids. Research on distantly related and much smaller-brained and bodied animals is of critical importance to understanding the extent to which different brains can generate common representations about the social world, as well as to identify phylogenetic and selective pressures that shape the evolution of these important cognitive mechanisms. I appreciate that the authors have developed a novel and naturalistic paradigm that closely mirrors work in primates and corvids. The paper is also succinct, highly readable, and engaging. Below I list several comments and suggestions to further improve the manuscript, which I believe should ultimately be published.

Discussion

The biggest question for me, which could be discussed further, is why male punishment does not depend on observing females and, critically, if it does not, why females should be sensitive to being observed. I appreciate that the authors' focus was on the female subjects and that these questions may be further probed in future experiments, but I think the manuscript would benefit from further discussion of them. With regard to the males, I am slightly confused by the cues that males could be picking up on. In study 1, they punish more according to the female's speed of cheating, regardless of whether they could actually see the cheating. In study 2, they punished more when the female went behind the opaque barrier, whether or not the males could actually see the barriers (and even though females did not cheat more quickly behind the opaque barrier). As the authors note, maybe the females are behaving differently or releasing different olfactory cues in these different contexts. The authors should definitely point to the need for future work to better understand what exactly males were picking up on and why the different contexts (i.e., invisible vs visible, cheating vs not cheating) led the females to produce different behavioral or chemical responses.

Females' behavior in block 1 of Study 2 suggests that they were not simply attracted to the opaque barrier (since they preferred the transparent barrier in the invisible condition). However, their quick change in strategy between blocks 1 and 2 raises questions that the authors could discuss: what other mechanisms are shaping females' strategies for maximizing intake while minimizing punishment? Does females' quick abandonment of the strategy that looks most like visual perspective-taking mean that this is not a dominant strategy among those they possess, that the experimental context cannot sustain such a strategy because it departs in meaningful ways from the natural contexts in which a perspective-taking strategy would be more reliably employed, or that the fish were relying on a mechanism besides perspective-taking all along? Flexibility is necessary for deception to be successful and it cannot be employed time and again in every instance (if I deceive every time, you will never trust me). However, the facts (1) that females might be giving themselves away and somehow eliciting punishment when they cheat, and (2) that punishment diminishes the use of the perspective-taking-like strategy also raises questions about whether punishment has really selected for the evolution of perspective-taking in the first place. But perhaps the authors would expect more consistent perspective-taking-like responses under even more naturalistic conditions (e.g., if females would not provide the punishment-eliciting cues, like going behind barriers or behaving in particular ways, under natural conditions, and ultimately would not be caught, punished, and deterred from the perspective-taking strategy)?

Finally, if males are able to detect female signals that shape their decision to punish, an important question is whether females may also be detecting signals (e.g., in behavior) that guide perspective-taking-like responses without any appreciation of perspective.

Ultimately, I think this study is excellent and should be published but I think further discussion of these issues would elevate its value and the impact it will have on future work.

Results

In study 2, was there variation in whether females switched tactics between block 1 and block 2? Was the decision to switch tactics or not between blocks 1 and 2 linked to male punishment behavior during block 1?

In the methods or supplement, it would be nice to see some data from the training session on rates of "cheating" and punishment, and perhaps whether such individual variation predicts performance in the tasks (or is this the data from which the metric of punishment was derived, that was then linked to the strategic cheating score in study 1?).

Introduction

Line 43-46: The authors may wish to begin this sentence: "While the exact mechanisms remain debated (see, e.g., Heyes, 2014; Martin and Santos, 2016; Krupenye and Call, 2019), these results and others..." I don't think it is necessary to extensively review these debates but it would be worthwhile to at least note their existence and point readers to relevant literature.

-Heyes, 2014: "Animal mindreading: What's the problem?"

-Martin and Santos, 2016: What cognitive representations support primate theory of mind?

-Krupenye and Call, 2019: "Theory of mind in animals: Current and future directions"

Clarification and further suggestion

Lines 152-153: is this measure of punishment an independent measure of each male's tendency to punish (e.g., from training) or is it their tendency to punish within the present experiment. If it is not independent, then there are questions about the causal direction of this relationship Are females more strategic when they have a more punitive partner, or are males more punitive in response to female behavior (e.g., if females with higher strategic cheating measures also cheated more quickly in general or were more likely to produce whatever cues males were apparently picking up on)? If the authors can show that strategic cheating score is not related to overall level of cheating across

conditions, this would provide further support for their inferred causal direction of the relationship between male punitiveness and female strategic cheating score.

Analysis Methods

Based on the description in the methods section, the authors have pursued a sound analysis approach. However, the way the analyses are presented in the main text sometimes gives the impression that the authors used a stepwise/sequential model building approach. Such a sequential approach is ill-advised not least because it creates multiple testing issues. The methods section makes clear that this was not the authors' approach. I would encourage them to make some adjustments to the description in the main text, to remove confusion earlier on. Specifically, descriptions like "Model fit was improved by the inclusion of X term" (e.g., line 179-181) give the impression that the models were built sequentially, which is not the case. In that instance, I would just say that the block term was not a significant predictor of X. In the methods, it is then important to specify how significance was determined (which the authors have already clearly done).

Overall, I would like to commend the authors on a novel, important, and well-written paper, and I look forward to seeing it published.

REVIEWER 1

Comment R1.1. This excellent manuscript describes an exciting study showing clear evidence for theory of mind in fishes. This is the first demonstration of this phenomenon outside of non-human primates and provides compelling evidence for a complex cognitive ability in cleaner fish. The study is very well executed and shows that paired male and female cleaner fish exhibit a complicated relationship where the female will cheat when the male is not watching her and will alter her behaviour when the males is visible. In response to cheating the male punishes the female and the degree of punishment modulates female cheating. In a second experiment female cleaner fish choose an opaque barrier to cheat behind rather than a transparent one so the male cannot see her cheating. In response to this it appears that the male punishes the female if she chooses the opaque barrier and so again this influences the future behavioural choices of the female. This is a clear demonstration of theory of mind where the female is aware of being watched and alters her behaviour accordingly. Decisions to punish females going out of site is a remarkable finding and suggests a complex relationship between the controlling male and the cheating female. These novel results are fascinating and are highly likely to spark media interest. I highly recommend publication after a few minor comments:

Response R1.1. We thank the reviewer for their positive evaluation of our MS and for their suggestions on how to improve it. We have addressed each of their comments in turn below.

Comment R1.2. L85 you mention study 1 but not study 2 in this paragraph - please amend.

Response R1.2. We were hoping to use this particular paragraph to provide an overview of Study 1 specifically. This allows us to introduce the partition manipulation without discussing the opaque and transparent barriers used in Study 2. Our thinking was that it would be helpful to focus on Study 1 initially and then use Study 1 to scaffold the explanation of our Study 2 design.

Comment R1.3. L247 More details are need for reproducibility reasons - what size were the animals? Presume you used natural seawater? Can you add how much water was replaced per day or week, aeration, the light dark regime, temperature and water quality parameters. Did you have ethical permission and what was the fate of the fish?

Response R1.3. Fish sizes are presented in Table S3 and we now mention this explicitly in the main text (p. 12). We did indeed use natural seawater and now mention this in our revision (p. 12). Specifically, our aquaria used fresh sea water and a 'flow through' system so water was constantly being replaced and thus matched the temperature of the surrounding seawater. Although we do not know the specific flow rate, our best guess is that it's approximately 1L/min. We used

natural light conditions. The aquaria were outdoors but covered to reduce sun exposure and were covered with nets when not in use to prevent bird predation. At the end of the field season, fish were returned to the reefs where they were initially caught. We had IACUC approval from both US institutions (Boston College and Yale University; Protocols 2015-11627 and 2016-006-01, respectively) and our project was approved by the French Polynesian authorities responsible for the program 'Délégation à la Recherche.' We have added this information under a new section entitled *Ethics*.

Comment R1.4.

SI: would it be possible to add some videos for study 1 and 2 here?

Response R1.4.

We have created a video for Studies 1 and 2 and we have uploaded it as supplementary materials.

REVIEWER 2

Comment R2.1. The authors present an exciting, well-designed, and highly novel study on the social cognitive abilities of a species of fish. As the authors note, little is known about the perspective-taking abilities of animals besides primates and corvids. Research on distantly related and much smaller-brained and bodied animals is of critical importance to understanding the extent to which different brains can generate common representations about the social world, as well as to identify phylogenetic and selective pressures that shape the evolution of these important cognitive mechanisms. I appreciate that the authors have developed a novel and naturalistic paradigm that closely mirrors work in primates and corvids. The paper is also succinct, highly readable, and engaging. Below I list several comments and suggestions to further improve the manuscript, which I believe should ultimately be published.

Response R2.1.

We thank the reviewer for their positive evaluation of our MS and for their suggestions on how to improve it. We have incorporated their suggested changes and conducted additional explorations of our data which we discuss here.

Comment R2.2. Discussion. The biggest question for me, which could be discussed further, is why male punishment does not depend on observing females and, critically, if it does not, why females should be sensitive to being observed. I appreciate that the authors' focus was on the female subjects and that these questions may be further probed in future experiments, but I think the manuscript would benefit from further discussion of them. With regard to the males, I am slightly confused by the cues that males could be picking up on. In study 1, they punish more according to the female's speed of cheating, regardless of whether they could actually see the cheating. In study 2, they punished more when the female went behind the opaque barrier, whether or not

the males could actually see the barriers (and even though females did not cheat more quickly behind the opaque barrier). As the authors note, maybe the females are behaving differently or releasing different olfactory cues in these different contexts. The authors should definitely point to the need for future work to better understand what exactly males were picking up on and why the different contexts (i.e., invisible vs visible, cheating vs not cheating) led the females to produce different behavioral or chemical responses.

Response R2.2. We agree that this is an important point and one which we address more directly in our revised discussion (p. 10-11).

Comment R2.3. Females' behavior in block 1 of Study 2 suggests that they were not simply attracted to the opaque barrier (since they preferred the transparent barrier in the invisible condition). However, their quick change in strategy between blocks 1 and 2 raises questions that the authors could discuss: what other mechanisms are shaping females' strategies for maximizing intake while minimizing punishment? Does females' quick abandonment of the strategy that looks most like visual perspective-taking mean that this is not a dominant strategy among those they possess, that the experimental context cannot sustain such a strategy because it departs in meaningful ways from the natural contexts in which a perspective-taking strategy would be more reliably employed, or that the fish were relying on a mechanism besides perspective-taking all along? Flexibility is necessary for deception to be successful and it cannot be employed time and again in every instance (if I deceive every time, you will never trust me). However, the facts (1) that females might be giving themselves away and somehow eliciting punishment when they cheat, and (2) that punishment diminishes the use of the perspective-taking-like strategy also raises questions about whether punishment has really selected for the evolution of perspective-taking in the first place. But perhaps the authors would expect more consistent perspective-taking-like responses under even more naturalistic conditions (e.g., if females would not provide the punishment-eliciting cues, like going behind barriers or behaving in particular ways, under natural conditions, and ultimately would not be caught, punished, and deterred from the perspective-taking strategy)?

We agree that females' change in strategy during Study 2 is intriguing. In our revised discussion, we have elaborated on females' strategy and discuss how our data relate to the selective forces at play under natural conditions (p. 11). The fact that females initially use a perspective-taking-like strategy suggests that this tendency may be their default strategy during these interactions. That female barrier choices varied with block also speaks to the flexibility of their behavior. Specifically, our results raise the possibility that at least some females may be changing their strategies across time, potentially as a response to male behavior. We found that males punished females more when they chose the opaque barrier, independently of condition (see revised discussion in response to Comment R2.2). We did not find a significant block x female choice interaction, suggesting the block

did not significantly moderate the effect of female choice on male punishment. However, in an exploratory analysis, we examined blocks separately and found a marginal effect for female choice on male punishment (LRT, $p = 0.08$) in the first but not the second block. One tentative takeaway from this pattern of results is that female choice may vary as a function of the feedback they get from males. Exploring the interplay between cues exhibited by females depending on their behavior, males' responses and females' shifting strategies presents an exciting avenue for future research, as we now mention in our discussion.

Comment R2.4. Finally, if males are able to detect female signals that shape their decision to punish, an important question is whether females may also be detecting signals (e.g., in behavior) that guide perspective-taking-like responses without any appreciation of perspective.

Response R2.4. This is an interesting point that warrants further work and we now mention this in our revised discussion (p. 10-11).

Comment R2.5. Ultimately, I think this study is excellent and should be published but I think further discussion of these issues would elevate its value and the impact it will have on future work.

Response R2.5. Thank you very much for your comments. In addressing them, we believe we have better explored our current findings and helped pave the way for future work in this area.

Comment R2.6. Results. In study 2, was there variation in whether females switched tactics between block 1 and block 2? Was the decision to switch tactics or not between blocks 1 and 2 linked to male punishment behavior during block 1?

Response R2.6. There was some variation in whether females switched tactics between the first and second blocks (Fig. 1, now Fig. S4 in the Supplementary Materials). Within Block 1, male punishment was marginally predicted by females' choices of the opaque vs. transparent barrier (LRT, $p = 0.08$) and this relationship was not seen in Block 2 (note that the interaction between block and choice was not significant). The question of whether the decision to switch tactics across blocks is dependent on punishment in Block 1 is an interesting one. However, based on our exploration of the data, there is no relationship between getting punished in Block 1 and choosing the transparent option in Block 2 (Pearson correlation = -0.18 , $p = 0.6$; Fig. 2). We have not added these additional analyses to the results because they are so exploratory in nature, but we will add them if the reviewer or editor believes they would be a useful addition. For now, we have added individual-level plots to the SOM in addition to making our data available.

Fig. 1. Barrier choice by block number and condition, faceted by cleaner fish pairs. This plot is now Fig. S4 in the SOM.

Fig. 2. Relationship between male punishment in Block 1 and females’ choices of the transparent barrier in Block 2.

Comment R2.7. In the methods or supplement, it would be nice to see some data from the training session on rates of “cheating” and punishment, and perhaps whether such individual variation predicts performance in the tasks (or is this the data from which the metric of punishment was derived, that was then linked to the strategic cheating score in study 1?).

Response R2.7. While we do not have the training data in shape to examine these effects (e.g., training sessions were not video recorded and reliability coded as were test sessions), we can examine stability of cheating and punishment by looking across our two studies. In doing so we see that punishment appears to be relatively stable: there is a significant correlation between punishment in Study 1 and punishment in Study 2 (Fig. 3A; Pearson correlation, $r = 0.72$, $p = 0.012$). By contrast, female flake eating does not show the same relationship across studies (Fig. 3B): while Fig. 3B suggests that cheating is weakly positively related across the two studies, this was not a significant correlation (Pearson correlation, $r = 0.48$, $p = 0.13$). Note, also, that females ate more flakes in Study 1 than in Study 2 which is

why the X and Y intercepts are on different scales. We have not added these analyses to the MS because they are so exploratory in nature but we have made our data available. In so far as this comment relates to Comment 2.9 regarding the stability of male punishment: please note that we have now been more tentative in our revised MS about the direction of causality between female strategic cheating and male punishment (see Response 2.9).

Fig. 3. Relationship between male punishment behavior in Study 1 and Study 2 (A; Pearson correlation, $r = 0.72$, $N = 11$, $p = 0.12$) as well as the relationship between

female flake eating in Study 1 and Study 2 (B; Pearson correlation, $r = .48$, $N = 11$, $p = 0.13$).

Comment R2.8. Introduction. Line 43-46: The authors may wish to begin this sentence: “While the exact mechanisms remain debated (see, e.g., Heyes, 2014; Martin and Santos, 2016; Krupenye and Call, 2019), these results and others...” I don’t think it is necessary to extensively review these debates but it would be worthwhile to at least note their existence and point readers to relevant literature.

-Heyes, 2014: “Animal mindreading: What’s the problem?”

-Martin and Santos, 2016: What cognitive representations support primate theory of mind?

-Krupenye and Call, 2019: “Theory of mind in animals: Current and future directions”

Response R2.8. We like this suggestion a lot, and have added in the suggested caveat and references.

Comment R2.9. Clarification and further suggestion. Lines 152-153: is this measure of punishment an independent measure of each male’s tendency to punish (e.g., from training) or is it their tendency to punish within the present experiment. If it is not independent, then there are questions about the causal direction of this relationship Are females more strategic when they have a more punitive partner, or are males more punitive in response to female behavior (e.g., if females with higher strategic cheating measures also cheated more quickly in general or were more likely to produce whatever cues males were apparently picking up on)? If the authors can show that strategic cheating score is not related to overall level of cheating across conditions, this would provide further support for their inferred causal direction of the relationship between male punitiveness and female strategic cheating score.

Response R2.9. This is a good point and one which we now address in our revised discussion. While we cannot be certain of the directionality, we believe our data are more consistent with the idea that females are more strategic when they have a more punitive partner. When we examined male punishment across our studies (see also Response 2.7) we found a significant correlation, suggesting that male punishment was consistent across our two studies. We did not see this same relationship when we examined female cheating across studies. That male punishment is at least somewhat stable aligns with past work pointing to stable differences between pairs (e.g., the size differential between males and females predicts punishment; Raihani et al., 2012, PRSB). This being said, we now acknowledge in the revised MS that we cannot be sure of the directionality.

Comment R2.10. Analysis Methods. Based on the description in the methods section, the authors have pursued a sound analysis approach. However, the way the analyses are presented in the main text sometimes gives the impression that the authors used a

stepwise/sequential model building approach. Such a sequential approach is ill-advised not least because it creates multiple testing issues. The methods section makes clear that this was not the authors' approach. I would encourage them to make some adjustments to the description in the main text, to remove confusion earlier on. Specifically, descriptions like "Model fit was improved by the inclusion of X term" (e.g., line 179-181) give the impression that the models were built sequentially, which is not the case. In that instance, I would just say that the block term was not a significant predictor of X. In the methods, it is then important to specify how significance was determined (which the authors have already clearly done).

Overall, I would like to commend the authors on a novel, important, and well-written paper, and I look forward to seeing it published.

Response R2.10. Thank you for this suggestion and for your encouraging comments about our MS. We have made this change to our descriptions of our results.

REVIEWERS' COMMENTS:

Reviewer #2 (Remarks to the Author):

I appreciate that the authors have taken all of my suggestions and those of the other reviewer on board. I feel that the manuscript is now further improved and ready for publication. The video is also immensely helpful for visualizing the experiments and greatly enriches the manuscript. I also like the extra plots.

I would suggest one minor change to the new abstract: line 21: change "exhibit ToM capacities akin to" to "exhibit foundations of ToM akin to."

Otherwise, well done on a really cool study!